# Breast cancer classification based on the integration of diagnostic algorithms for calcifications and masses using a mixture of experts

Yuma Konaka[1], Takuya Ueda[1], Ryusei Inamori[2], Jumpei Sato[1], Keisuke Sugawara[1], Yuta Shiratori[2], Fumihito Hario[2], Eichi Takaya[2,3]*, Tomoya Kobayashi[2,3], Yoshikazu Okamoto[2,3]

1 Department of Diagnostic Radiology, Tohoku University Graduate School of Medicine, 2-1 Seiryo-Machi, Aoba-ku, Sendai, Miyagi, Japan, 2 Department of Diagnostic Imaging, Tohoku University Graduate School of Medicine, 2-1 Seiryo-Machi, Aoba-ku, Sendai, Miyagi, Japan, 3 AI Lab, Tohoku University Hospital, 1-1 Seiryo-machi, Aoba-ku, Sendai, Miyagi, Japan

* eichi.takaya.d5@tohoku.ac.jp

## Abstract

### Purpose

To investigate the effectiveness of an integrated deep-learning (DL) algorithm, the *Mixture of Radiological Findings Specific Experts (MoRFSE)*, in breast cancer classification by imitating the diagnostic decision-making process of radiologists.

### Methods

A total of 2,764 mammographic images (1,462 breast cancer, 248 benign lesions, and 1,054 normal breast tissue) from the TOMPEI-CMMD were used. The *MoRFSE* comprises three DL models: a gate network for categorization (gNet) and two classification expert networks (cExp and mExp) specialized in capturing the distinct characteristics of calcifications and masses, respectively. This structure imitates radiologists' comprehensive diagnostic process by applying distinct algorithms for different lesion types. The classification performance of *MoRFSE* was compared with the *Conventional ResNet18* model. 5-fold cross-validation was used, and performance was assessed using the area under the receiver operating characteristic curve (AUC). DeLong's test was performed to evaluate statistical significance.

### Results

*MoRFSE* achieved a significantly higher AUC (0.9616) compared to *Conventional ResNet18* (0.9577, p = 0.001348).

**Data availability statement:** The TOMPEI-CMMD used in our study is available as open data via The Cancer Imaging Archive (TCIA) online data repository: https://doi.org/10.7937/WEZW-BH22.

**Funding:** This work was supported by the Japan Science and Technology (JST) Agency, as core research for evolutional science and technology (CREST) Grant No. JPMJCR15D1.

**Competing interests:** The authors have declared that no competing interests exist.

## Conclusion

By integrating specialized algorithms for calcifications and masses, the *MoRFSE* model effectively emulates radiologists' diagnostic process. These findings suggest that *MoRFSE* has the potential to improve the accuracy of breast cancer diagnosis based on mammograms.

## Introduction

Breast cancer remains the most commonly diagnosed and lethal cancer among women worldwide, with both its incidence and mortality rates on the rise [1,2]. Mammography screening has been used for the early detection of breast cancer, reducing mortality by approximately 20% in asymptomatic women [3,4]. Despite these benefits, mammography screening has its limitations [5].

When interpreting mammograms, radiologists focus on identifying abnormal breast structures such as calcifications and masses, which requires a distinct assessment [6]. For calcifications, radiologists evaluate the shape of each calcification site and the overall distribution pattern of calcifications within the breast. When evaluating a mass, the focus shifts to analyzing its shape and margins. This approach underscores the need for distinct diagnostic algorithms for each lesion type to ensure more comprehensive and accurate evaluations.

In recent years, artificial intelligence (AI)-based classification systems, especially those utilizing convolutional neural networks (CNN), a type of deep learning (DL) algorithms, have made significant advances in breast cancer classification [7,8]. Although some DL-based breast cancer classification systems have demonstrated their usefulness in clinical applications, they still face limitations in their classification performance [9,10].

Many existing DL models process mammographic findings such as calcifications and masses without explicitly distinguishing between them [11]. From our perspective, this lack of lesion-specific processing may contribute to limitations in diagnostic accuracy, as it diverges from the interpretive strategies of radiologists, who assess these lesion types using different criteria. We hypothesize that treating fundamentally different findings with a single, undifferentiated algorithm may hinder the model's ability to fully capture their distinct characteristics.

To address this issue, some studies have developed CNN-based approaches that focus on a single lesion type. For example, Chakravarthy et al. [12] focused on the detection and classification of microcalcifications, while Barnett et al. [13] developed an interpretable model targeting mass lesions. These studies highlight the benefits of radiological finding-specific modeling, yet they are limited in that each model is optimized for only one type of lesion, and thus cannot generalize to the other.

The Mixture of Experts (MoE) model is a neural network architecture that addresses complex problems by dividing them into simpler sub-problems [14,15]. This model leverages multiple specialized networks, referred to as "experts," each designed to solve a specific task, together with a "gate" network that dynamically

allocates the contribution of each expert to the final output [14,15]. By harnessing the diverse expertise of these networks, the MoE model aims to enhance comprehensive decision-making capabilities and solve complex problems that conventional singular architectures are unable to address.

The MoE model is widely used in large language models (LLMs) and has been applied across diverse fields such as natural language processing [16–18], computer vision [19–21], and multimodality [22–24]. In recent years, the application of MoE in medical research has gained increasing attention and demonstrated promising potential [25–28]; however, its application to medical image classification, particularly in breast cancer, remains limited.

Leveraging the MoE model in this context presents significant opportunities to bridge the gap between the capabilities of current DL systems and the diagnostic strategies employed by radiologists. By incorporating radiological finding-specific algorithms within a unified framework, the MoE model may enable diagnostic strategies that better align with radiologists' approaches.

The purpose of our study is to investigate the application of the MoE model to breast cancer diagnosis, focusing on its potential to imitate the detailed and radiological finding-specific decision-making processes of radiologists.

## Materials and methods

### Dataset

Fig 1 shows the data selection process. Our study utilized the TOMPEI-CMMD [29], a refined version of the Chinese Mammography Database (CMMD) [30]. This dataset contains 2,436 mediolateral oblique (MLO) views of mammographic images from 1,775 Chinese patients who underwent mammography examinations between July 2012 and January 2016. A board-certified radiologist with over 20 years of experience in breast cancer diagnosis assessed these images with prior knowledge of the diagnostic labels provided in the original CMMD dataset. For each lesion, information on malignancy (breast cancer or benign lesion), radiological findings (calcification, mass, focal asymmetric density (FAD), or distortion), and location was documented. Segmentation masks were generated based on these evaluations, and images with multiple findings were assigned multiple tags.

A total of 2,764 breast images were employed for our investigation, including 1,462 images of breast cancer, 248 images of benign lesions and 1,054 images of normal breast tissue.

From the 2,436 breast images, regions containing breast cancer or benign lesions were cropped to 512×512 pixels, centered around the segmentation masks. If multiple lesions were present in a single image, each lesion was individually extracted. In our study, those tagged as distortion were excluded. For breasts with normal breast tissue, the mammogram was binarized to differentiate the breast area from the background, and random 512×512 pixel regions were selected

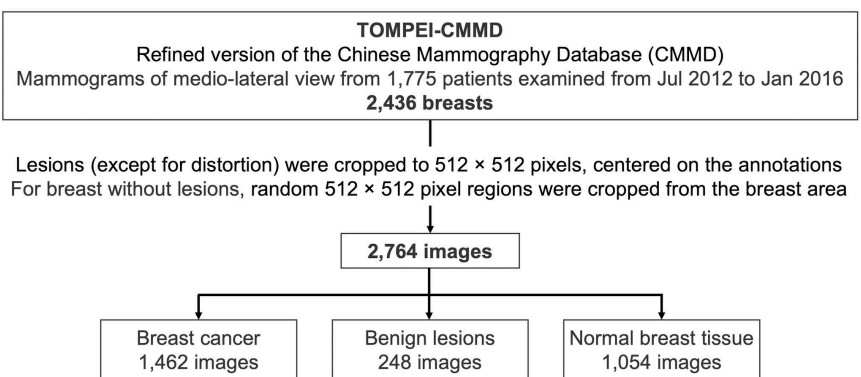

**Fig 1. Flowchart of inclusion and exclusion criteria for breast images in our study.**

within the breast area. As a result, our study utilized a total of 2,764 images for analysis: 1,462 images of breast cancer, 248 images of benign lesions, and 1,054 images of normal breast tissue. Based on the annotations, two types of labels were assigned to each image: a *diagnostic label* indicating the malignancy (breast cancer, benign lesions, or normal breast tissue), and a *radiological finding label* specifying lesion characteristics (calcification or mass). Lesions tagged as FAD were classified as mass.

**Mixture of Radiological Findings Specific Experts (MoRFSE).** Fig 2 illustrates the structure of *Mixture of Radiological Findings Specific Experts (MoRFSE)* conceived in our study. *MoRFSE* consists of three DL models: one gate network for categorization (gNet) and two classification expert networks specialized in the distinct characteristics of calcifications and masses, referred as the calcification expert (cExp) and the mass expert (mExp), respectively. The gNet evaluates the radiological features of the breast to determine whether the lesions are primarily characterized by calcifications or masses. Then, the cExp and mExp independently assess the presence of malignancy to predict whether the image indicates breast cancer, benign lesion, or normal breast tissue. The prediction probabilities from cExp and mExp were independently multiplied by the probabilities of the radiological feature type predicted by gNet. These weighted probabilities were then summed to yield the final output, denoting the probabilities of breast cancer, benign lesions, or normal breast tissue. This approach refines the prediction by integrating the distinct features observed from calcification-specific and mass-specific perspectives on mammograms.

*MoRFSE* model consists of three deep learning models. Prediction probabilities from cExp and mExp were weighted by the corresponding probabilities from gNet; then summed to yield the final decision output.

*MoRFSE* is expressed by the following equation, where $x$ represents the input and $y$ represents the final output.

$$y = w_c \times Expert_c(x) + w_m \times Expert_m(x)$$

where, $w_c$ and $w_m$ are the probabilities of calcifications and masses predicted by gNet, respectively, and $Expert_c(x)$ and $Expert_m(x)$ represent the probabilities of breast cancer, benign lesions, or normal breast tissue predicted by cExp and mExp. To be able to interpret $y$ as a probability, we set $w_c + w_m = 1$.

In our study, three individual ResNet18 architectures [31] were applied to each model (gNet, cExp, and mExp).

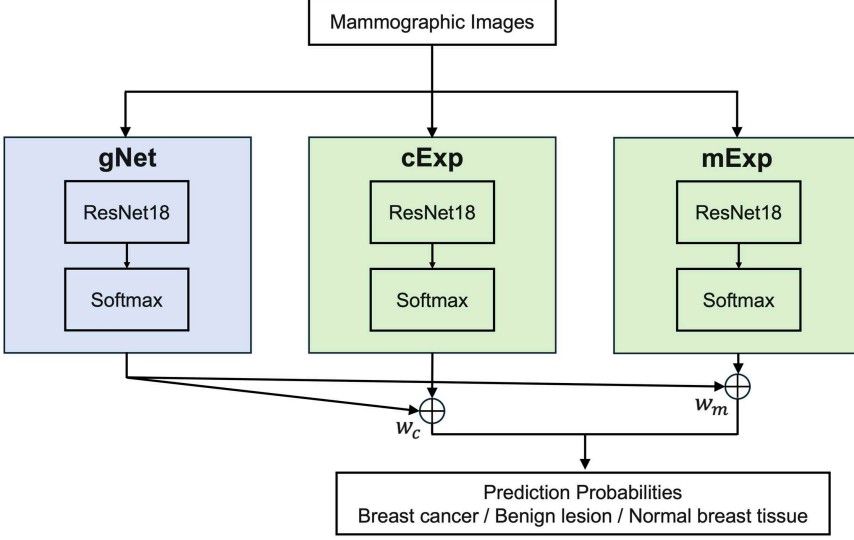

**Fig 2. Architecture of the *MoRFSE*.**

## Comparison methods

To evaluate the effectiveness of *MoRFSE*, we compared its performance with two comparison models: *Conventional ResNet18* and *No-gNet-MoRFSE*. *Conventional ResNet18* is a model that employs a single CNN to evaluate malignancy on each image without differentiating between calcifications and masses.

*No-gNet-MoRFSE* is a version of *MoRFSE* where the gNet is omitted, functioning as an ensemble method that averages the outputs of multiple models. This model applies equal weights ($w_c = w_m = 0.5$) to combine the predictions of cExp and mExp, regardless of the radiological characteristics of the input image.

Both models generated predictions for three classification categories: breast cancer, benign lesions, and normal breast tissue.

## Evaluations

In our study, 5-fold cross-validation was used to evaluate the generalization performance of the models. The given dataset was initially divided into five mutually exclusive folds. Each fold was used as validation data once, while the remaining four folds served as training data. This cycle of training and validation was repeated for each of the five folds, ensuring that every fold was used as validation data once. This entire cycle of training and validation across the five folds was repeated five times, and then the mean prediction for each image was computed. Finally, the mean out-of-fold (OOF) predictions and corresponding true values for each fold were integrated to create an evaluation dataset that reflected the consistent predictive performance of the model for all data. The OOF prediction and true value pairs were used to calculate the area under the receiver operating characteristic curve (AUC) to evaluate the classification accuracy of distinguishing breast cancer.

To provide a more detailed evaluation, the AUC evaluating the classification accuracy of distinguishing breast cancer was also assessed for each radiological finding (calcifications and masses) using the same cross-validation procedure. This evaluation served as a supplementary analysis to the main analysis and was intended to elucidate the characteristics and trends of the model from a medical perspective.

## Statistical analysis

DeLong's test was applied to evaluate statistical significance. The entire OOF prediction data obtained through 5-fold cross-validation was considered as a single test set, and DeLong's test was performed on this integrated data. A p-value of less than 0.05 was considered statistically significant.

In the supplementary analysis of each radiological finding, statistical testing was not applied. This is because the analysis was not intended for direct statistical comparison between models or lesion types. Instead, it aimed to descriptively explore the behavior and characteristics of the models when applied to calcification- and mass-specific lesions, offering insights into the model's diagnostic tendencies from a radiological perspective.

## Implementation

The gNet in *MoRFSE* was trained exclusively on images containing calcifications or masses, using *radiological finding labels* to categorize the lesion type. The cExp was trained on images containing calcification or normal breast tissue, using *diagnostic labels* to classify the images as breast cancer, benign lesions, or normal breast tissue. Similarly, the mExp was trained on images containing masses or normal breast tissue, also utilizing *diagnostic labels* for the same classification task.

For the *conventional ResNet18*, all images with *diagnostic labels* in the training dataset were used without distinguishing between radiological findings. In the *No-gNet-MoRFSE* model, the cExp and mExp were utilized with fixed equal weights ($w_c = w_m = 0.5$) to combine their predictions, bypassing the functionality of gNet.

The weights of each network were initialized using a pre-trained model on ImageNet [32]. In our study, the training parameters were set with a batch size of 64, and training was conducted for 30 epochs. Adaptive moment estimation (Adam) algorithm [33] was used for optimization (learning rate = 1e-5, weight decay = 0), and cross entropy was used as the loss function.

Codes available at https://github.com/eichitakaya/MoRFSE.

## Computing environment

The computing system used in our study consisted of AMD Ryzen Threadripper Pro 3955WX with 16 cores (Advanced Micro Devices, Inc., Santa Clara, CA, USA) and NVIDIA RTX A6000 GPU with 48 GB of memory (Nvidia Co., Santa Clara, CA, USA). The operating system was Ubuntu 20.04.5 long term support (LTS), and the DL models were implemented using Python 3.7.3 with PyTorch 1.13.1.

## Results

The AUCs of *MoRFSE*, *Conventional ResNet18* and *No-gNet-MoRFSE* using 5-fold cross-validation are shown in Table 1. The AUC of *MoRFSE* was significantly higher than that of *Conventional ResNet18* (p = 0.001348), with AUCs of 0.9616 and 0.9577, and was significantly higher than that of *No-gNet-MoRFSE* (p < 0.001), with AUCs of 0.9616 and 0.9561. The AUC of *Conventional ResNet18* was higher than that of *No-gNet-MoRFSE* (p = 0.8476), with AUCs of 0.9577 and 0.9561. Statistical analysis revealed no significant differences between *Conventional ResNet18* and *No-gNet-MoRFSE*.

The AUCs for each radiological finding are shown in Table 2. The Calcification-AUC of *MoRFSE* was higher than that of *Conventional ResNet18* and *No-gNet-MoRFSE*, with AUCs of 0.8448, 0.8446 and 0.8152, respectively. The Mass-AUC of *MoRFSE* was higher than that of *Conventional ResNet18* and *No-gNet-MoRFSE*, with AUCs of 0.8389, 0.8323 and 0.8283, respectively.

**Table 1. The AUC for *MoRFSE*, *Conventional ResNet18* and *No-gNet-MoRFSE*.**

| Model | AUC |
|---|---|
| *MoRFSE* | **0.9616 (0.9552-0.9681)** |
| *Conventional ResNet18* | 0.9577 (0.9507-0.9646) |
| *No-gNet-MoRFSE* | 0.9561 (0.9490-0.9632) |

Note: The **bold font** indicates the highest score among the three models. Values in parentheses represent the 95% confidence interval.

**Table 2. The AUC for each radiological finding for *MoRFSE*, *Conventional ResNet18* and *No-gNet-MoRFSE*.**

| Model | Calcification-AUC | Mass-AUC |
|---|---|---|
| *MoRFSE* | **0.8389 (0.7934-0.8845)** | **0.8448 (0.8152-0.8743)** |
| *Conventional ResNet18* | 0.8323 (0.7862-0.8784) | 0.8446 (0.8143-0.8749) |
| *No-gNet-MoRFSE* | 0.8283 (0.7855-0.8712) | 0.8152 (0.7826-0.8478) |

Note: The **bold font** indicates the highest score among the three models. Values in parentheses represent the 95% confidence interval.

## Discussion

In our study, *MoRFSE* achieved a significantly higher AUC than *Conventional ResNet18*. This finding suggests that the architecture of *MoRFSE*, which imitates the radiologists' clinical diagnostic approach through the MoE framework, may lead to improved diagnostic accuracy. Specifically, the gNet integrates individual predictions generated by the two expert networks (cExp and mExp), enabling the effective application of radiological finding-specific algorithms. Unlike *Conventional ResNet18*, which relies on a single network, *MoRFSE* utilizes specialized networks to model unique lesion characteristics more accurately. This approach aligns with prior research by Shimokawa et al., indicating that incorporating radiologists' diagnostic strategies into DL models enhances performance [34]. These findings underscore the potential of leveraging domain-specific insights to develop more precise DL models, ultimately facilitating better clinical decision-making for breast cancer patients.

Although the performance difference between *MoRFSE* and *Conventional ResNet18* was statistically significant, the absolute AUC improvement (0.9616 vs. 0.9577) may appear marginal in the context of clinical decision-making. In typical clinical practice, especially under a 5% error tolerance, such incremental gains may not immediately translate into tangible improvements in diagnostic outcomes. However, even minor improvements in classification performance can have meaningful implications when scaled across large screening populations, potentially reducing false-positive or false-negative cases. Furthermore, the *MoRFSE* model contributes beyond raw performance metrics by offering a more interpretable and radiologist-aligned framework, which may facilitate greater clinical acceptance and trust in AI-based tools.

A closer examination of performance by lesion type revealed that *MoRFSE* achieved higher AUCs for both calcification-dominant and mass-forming lesions compared to *Conventional ResNet18*. This trend suggests that independently modeling lesion types and effectively combining their outputs may contribute to improved classification accuracy. Common DL models for breast cancer classification typically rely on a single network trained with a single label, such as breast cancer, benign lesions, or normal breast tissue. We hypothesize that such models may rely on average characteristic differences between calcification-associated and mass-associated breast cancers during classification, potentially limiting their ability to capture unique features specific to each lesion type. This limitation could lead to misdiagnosis by failing to identify the characteristics unique to calcifications or masses, thereby increasing false negatives and false positives in breast classification. While some studies have focused exclusively on either calcifications or masses [12,13,35,36], these approaches lack the ability to generalize effectively across different lesion types. In contrast, radiologists adapt their diagnostic strategies based on lesion type [6], analyzing specific features such as the individual shape and distribution patterns of the calcifications or the shapes and margins of the masses. These independent assessments are then integrated into a comprehensive evaluation to determine the presence or absence of breast cancer. *MoRFSE* effectively imitates this radiological approach by employing the gNet to integrate radiological finding-specific predictions, enabling the model to capture unique characteristics more effectively.

When comparing *MoRFSE* with *No-gNet-MoRFSE*, *MoRFSE* achieved a significantly higher AUC, suggesting the importance of the gNet component. The gNet dynamically weights the contributions of the expert networks (cExp and mExp) based on input features, effectively leveraging the strengths of these specialized networks. In contrast, *No-gNet-MoRFSE* lacks this weighting mechanism and relies on fixed equal weighting ($w_c = w_m = 0.5$), which prevents it from prioritizing the predictions of the most relevant expert network. As a result, *No-gNet-MoRFSE* is more susceptible to the influence of inaccurate predictions from the less specialized network, diminishing its overall performance.

A comparison of *Conventional ResNet18* with *No-gNet-MoRFSE* suggests the importance of dynamic weighting. Although *No-gNet-MoRFSE* integrates predictions from two specialized expert networks, its fixed weighting mechanism diminishes the potential advantage of specialization, resulting in a lower AUC than *Conventional ResNet18*. In contrast, *Conventional ResNet18* uses a single network trained on all lesion types without distinguishing between calcifications and masses. This training approach may have contributed to its consistent performance across diverse input types, albeit

without leveraging the specialization offered by the expert networks. The absence of a dynamic weighting mechanism in *No-gNet-MoRFSE* likely explains why its performance did not surpass that of the *Conventional ResNet18* model.

Our study has several limitations. First, our study was conducted using a single dataset, which limited the ability to make direct comparisons with previous studies and may affect the generalizability of the findings. To validate the performance of *MoRFSE* in a broader and comparative context, future studies using publicly available benchmark datasets that encompass diverse populations and imaging conditions are warranted. Second, the model's performance was evaluated using a single set of hyperparameters. To investigate the effectiveness and robustness of *MoRFSE*, further studies with various hyperparameter configurations are necessary. Finally, the limited dataset size may have affected the statistical significance of the results. Expanding the dataset size and diversity could address this limitation and provide more definitive evidence for the proposed model's advantages.

## Conclusion

In our study, we proposed *MoRFSE*, a deep learning framework that integrates specialized algorithms for calcifications and masses to better mimic radiologists' diagnostic strategies. *MoRFSE* achieved significantly higher classification performance compared to conventional models, indicating the effectiveness of radiological finding-specific expert networks and a dynamic gate mechanism. Our results suggest that incorporating domain-specific knowledge into model architecture can improve diagnostic accuracy in mammography-based breast cancer detection. Future work should aim to validate these findings across diverse datasets and investigate the potential integration of additional radiological findings such as architectural distortion.

## Author contributions

**Conceptualization:** Yuma Konaka, Takuya Ueda, Eichi Takaya, Tomoya Kobayashi.

**Data curation:** Yuma Konaka, Ryusei Inamori, Jumpei Sato, Keisuke Sugawara, Yuta Shiratori, Fumihito Hario.

**Formal analysis:** Yuma Konaka.

**Funding acquisition:** Takuya Ueda.

**Investigation:** Yuma Konaka.

**Methodology:** Yuma Konaka, Eichi Takaya.

**Project administration:** Yuma Konaka.

**Resources:** Takuya Ueda.

**Software:** Yuma Konaka, Eichi Takaya.

**Supervision:** Takuya Ueda, Yoshikazu Okamoto.

**Validation:** Yuma Konaka.

**Visualization:** Yuma Konaka.

**Writing – original draft:** Yuma Konaka.

**Writing – review & editing:** Takuya Ueda, Eichi Takaya, Tomoya Kobayashi, Yoshikazu Okamoto.

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
