## [Decision Letter · Decision Letter 0]

15 Jul 2025

PONE-D-25-25798Breast cancer classification based on the integration of diagnostic algorithms for calcifications and masses using a mixture of expertsPLOS ONE

Dear Dr. Takaya,

Thank you for submitting your manuscript to PLOS ONE. After careful consideration, we feel that it has merit but does not fully meet PLOS ONE’s publication criteria as it currently stands. Therefore, we invite you to submit a revised version of the manuscript that addresses the points raised during the review process.

 Please submit your revised manuscript by Aug 29 2025 11:59PM. If you will need more time than this to complete your revisions, please reply to this message or contact the journal office at plosone@plos.org . Please include the following items when submitting your revised manuscript:

We look forward to receiving your revised manuscript.

Kind regards,

Fahad Farhan Almutairi, PhD

Academic Editor

PLOS ONE

**Journal Requirements:**

1. When submitting your revision, we need you to address these additional requirements. Please ensure that your manuscript meets PLOS ONE's style requirements, including those for file naming. The PLOS ONE style templates can be found at https://journals.plos.org/plosone/s/file?id=wjVg/PLOSOne_formatting_sample_main_body.pdf and https://journals.plos.org/plosone/s/file?id=ba62/PLOSOne_formatting_sample_title_authors_affiliations.pdf 2. Please note that PLOS ONE has specific guidelines on code sharing for submissions in which author-generated code underpins the findings in the manuscript. In these cases, we expect all author-generated code to be made available without restrictions upon publication of the work. Please review our guidelines at https://journals.plos.org/plosone/s/materials-and-software-sharing#loc-sharing-code and ensure that your code is shared in a way that follows best practice and facilitates reproducibility and reuse. 3. Thank you for stating in your Funding Statement: This work was supported by the Japan Science and Technology (JST) Agency, as core research for evolutional science and technology (CREST) Grant No. JPMJCR15D1.  Please provide an amended statement that declares *all* the funding or sources of support (whether external or internal to your organization) received during this study, as detailed online in our guide for authors at http://journals.plos.org/plosone/s/submit-now.  Please also include the statement “There was no additional external funding received for this study.” in your updated Funding Statement. Please include your amended Funding Statement within your cover letter. We will change the online submission form on your behalf.

Reviewers' comments:

Reviewer's Responses to Questions

**Comments to the Author**

1. Is the manuscript technically sound, and do the data support the conclusions?

Reviewer #1: Yes

Reviewer #2: Yes

2. Has the statistical analysis been performed appropriately and rigorously? 

Reviewer #1: Yes

Reviewer #2: Yes

3. Have the authors made all data underlying the findings in their manuscript fully available?

Reviewer #1: Yes

Reviewer #2: Yes

4. Is the manuscript presented in an intelligible fashion and written in standard English?

Reviewer #1: Yes

Reviewer #2: Yes

5. Review Comments to the Author

**Reviewer #1: ** This study presents a well-executed investigation into the application of Mixture of Experts for mammographic classification, rigorously adhering to methodological standards. The manuscript is clearly structured, and the experimental design - including dataset curation, model architecture, and statistical validation - is transparent and reproducible.

While the field of medical imaging AI is saturated with incremental contributions, this work stands out by meticulously addressing methodological nuances often overlooked in similar studies. The integration of radiological domain knowledge into the MoE framework enriches the scientific discourse, offering a replicable template for future research.

Minor feedback for the authors’ consideration:

The reported performance gain is statistically significant but biologically marginal, given the conventional 5% error tolerance in clinical settings. While this does not diminish the study’s technical rigor, it invites reflection on whether such differences translate to tangible clinical utility. That said, the result remains an unequivocal scientific fact, and I commend the authors for their transparency in reporting it.

Overall, this is a valuable contribution to the field, and I appreciate the opportunity to review this work.

**Reviewer #2:**  The article was reviewed under the title " Breast cancer classification based on the integration of diagnostic algorithms for calcifications and masses using a mixture of experts".

Overall, the focus of the study is good. However, I would like to offer several recommendations that authors may find useful in the process of revising their manuscript:

1- Given the importance of the topic, it is necessary to include a review of previous studies in the article.

2- The phrase " in the supplementary analysis of each radiological finding, statistical testing was not applied, as the purpose was not formal comparison but rather descriptive exploration of model behavior " is used in the text of the article. Please provide a more detailed description of this phrase.

3- Abbreviations in the article are given in a paragraph titled "Abbreviations". It is better to introduce them in parentheses in the text of the article for the first time, and then only use the abbreviations.

4- Considering the results presented for the evaluation parameters, it is better to compare these results with the results presented by other researchers in other articles.

5- The conclusion section is very brief and needs to be rewritten.

6. PLOS authors have the option to publish the peer review history of their article (what does this mean? ). If published, this will include your full peer review and any attached files.

**Do you want your identity to be public for this peer review?** For information about this choice, including consent withdrawal, please see our Privacy Policy .

Reviewer #1: **Yes: ** Artyom Borbat

Reviewer #2: No

---

## [Author Response · Author response to Decision Letter 1]

31 Jul 2025

Response to Reviewers

Reviewer #1

1. The reported performance gain is statistically significant but biologically marginal, given the conventional 5% error tolerance in clinical settings. While this does not diminish the study's technical rigor, it invites reflection on whether such differences translate to tangible clinical utility. That said, the result remains an unequivocal scientific fact, and I commend the authors for their transparency in reporting it.

Response:

We sincerely appreciate the reviewer’s thoughtful observation and kind words regarding the scientific rigor of our study. We fully agree that while the performance improvement of MoRFSE over the conventional ResNet18 model was statistically significant, the absolute AUC difference (0.9616 vs. 0.9577) may seem marginal in the context of clinical error tolerance. In response to this valuable comment, we have added a discussion in the revised manuscript to explicitly address this point and to reflect on its clinical implications.

Added to Discussion (pages 10 to 11, lines 252 to 259):

“Although the performance difference between MoRFSE and Conventional ResNet18 was statistically significant, the absolute AUC improvement (0.9616 vs. 0.9577) may appear marginal in the context of clinical decision-making. In typical clinical practice, especially under a 5% error tolerance, such incremental gains may not immediately translate into tangible improvements in diagnostic outcomes. However, even minor improvements in classification performance can have meaningful implications when scaled across large screening populations, potentially reducing false-positive or false-negative cases. Furthermore, the MoRFSE model contributes beyond raw performance metrics by offering a more interpretable and radiologist-aligned framework, which may facilitate greater clinical acceptance and trust in AI-based tools.”

We hope this addition appropriately addresses the reviewer’s concern and provides a more balanced interpretation of the results from both statistical and clinical perspectives.

Reviewer #2

1. Given the importance of the topic, it is necessary to include a review of previous studies in the article.

Response:

We thank the reviewer for pointing out the need to include a more thorough review of related work, which we agree is essential to contextualize our contribution. In response, we have revised the Introduction section to incorporate a discussion of previous deep learning (DL) approaches that focused on lesion-specific modeling in mammography.

Added to Introduction (page 3, lines 64 to 74):

“Many existing DL models process mammographic findings such as calcifications and masses without explicitly distinguishing between them [11]. From our perspective, this lack of lesion-specific processing may contribute to limitations in diagnostic accuracy, as it diverges from the interpretive strategies of radiologists, who assess these lesion types using different criteria. We hypothesize that treating fundamentally different findings with a single, undifferentiated algorithm may hinder the model’s ability to fully capture their distinct characteristics.

To address this issue, some studies have developed CNN-based approaches that focus on a single lesion type. For example, Chakravarthy et al. [12] focused on the detection and classification of microcalcifications, while Barnett et al. [13] developed an interpretable model targeting mass lesions. These studies highlight the benefits of radiological finding-specific modeling, yet they are limited in that each model is optimized for only one type of lesion, and thus cannot generalize to the other.”

We believe this addition strengthens the manuscript by better situating our approach within the existing literature and clarifying how our work builds upon and extends prior research efforts.

2. The phrase "in the supplementary analysis of each radiological finding, statistical testing was not applied, as the purpose was not formal comparison but rather descriptive exploration of model behavior" is used in the text of the article. Please provide a more detailed description of this phrase.

Response:

We appreciate the reviewer’s comment pointing out the need for greater clarity in describing the purpose and rationale of the supplementary analysis. In response, we have revised the corresponding section in the Materials and methods / Statistical analysis to provide a more detailed and precise explanation of the intent behind the analysis and the reason for omitting statistical testing.

Added to Materials and methods / Statistical analysis (page 7 to 8, lines 183 to 186):

“In the supplementary analysis of each radiological finding, statistical testing was not applied. This is because the analysis was not intended for direct statistical comparison between models or lesion types. Instead, it aimed to descriptively explore the behavior and characteristics of the models when applied to calcification- and mass-specific lesions, offering insights into the model’s diagnostic tendencies from a radiological perspective.”

We hope this revision adequately clarifies the purpose and scope of the supplementary analysis and addresses the reviewer’s concern.

3. Abbreviations in the article are given in a paragraph titled "Abbreviations". It is better to introduce them in parentheses in the text of the article for the first time, and then only use the abbreviations.

Response:

We thank the reviewer for the helpful suggestion regarding the use of abbreviations. In accordance with the reviewer’s recommendation, we revised the manuscript to introduce each abbreviation at its first occurrence in the main text, followed by the abbreviated form in parentheses. Thereafter, only the abbreviations are used throughout the manuscript.

In particular, we removed the list of abbreviations from the caption of Figure 2 (page 6, lines 138 to 140) and revised the corresponding text in the Materials and methods / Mixture of Radiological Findings Specific Experts (MoRFSE) section (page 5, lines 126 to 129) to define all abbreviations appropriately upon their first mention.

We believe this change improves the readability and clarity of the manuscript and appreciate the reviewer’s attention to this detail.

4. Considering the results presented for the evaluation parameters, it is better to compare these results with the results presented by other researchers in other articles.

Response:

We appreciate the reviewer’s suggestion to compare our evaluation results with those from previous studies. However, as our study was conducted using a proprietary dataset not commonly used in the existing literature, there were no directly comparable studies available. To address this limitation, we have added a statement to the Discussion section to acknowledge this issue and to highlight the need for future research using publicly available benchmark datasets.

Added to Discussion (page 12, lines 295 to 298):

“First, our study was conducted using a single dataset, which limited the ability to make direct comparisons with previous studies and may affect the generalizability of the findings. To validate the performance of MoRFSE in a broader and comparative context, future studies using publicly available benchmark datasets that encompass diverse populations and imaging conditions are warranted.”

We hope this revision clarifies the scope of our current work and appropriately addresses the reviewer’s concern regarding comparative evaluation.

5. The conclusion section is very brief and needs to be rewritten.

Response:

We thank the reviewer for this valuable feedback. In response, we have substantially revised the Conclusion section to provide a more comprehensive summary of our study's contributions, findings, and future directions. The new conclusion emphasizes the significance of our proposed method, its alignment with radiologists’ diagnostic processes, and the potential for broader application.

Revised Conclusion (pages 12 to 13, lines 307 to 313):

“In our study, we proposed MoRFSE, a deep learning framework that integrates specialized algorithms for calcifications and masses to better mimic radiologists’ diagnostic strategies. MoRFSE achieved significantly higher classification performance compared to conventional models, indicating the effectiveness of radiological finding-specific expert networks and a dynamic gate mechanism. Our results suggest that incorporating domain-specific knowledge into model architecture can improve diagnostic accuracy in mammography-based breast cancer detection. Future work should aim to validate these findings across diverse datasets and investigate the potential integration of additional radiological findings such as architectural distortion.”

We believe this revision enhances the clarity and impact of the conclusion and better reflects the significance of our work.

---

## [Editor Report · Decision Letter 1]

10 Aug 2025

Breast cancer classification based on the integration of diagnostic algorithms for calcifications and masses using a mixture of experts

PONE-D-25-25798R1

Dear Dr. Takaya,

We’re pleased to inform you that your manuscript has been judged scientifically suitable for publication and will be formally accepted for publication once it meets all outstanding technical requirements.

Kind regards,

Fahad Farhan Almutairi, PhD

Academic Editor

PLOS ONE

---

## [Editor Report · Acceptance letter]

PONE-D-25-25798R1

PLOS ONE

Dear Dr. Takaya,

I'm pleased to inform you that your manuscript has been deemed suitable for publication in PLOS ONE. Congratulations! Your manuscript is now being handed over to our production team.

Kind regards,

on behalf of

Dr. Fahad Farhan Almutairi

Academic Editor

PLOS ONE